# Physiological Responses during Parental Conflicts as Potential Biomarkers for Adolescent Depression

**DOI:** 10.3390/children10071195

**Published:** 2023-07-10

**Authors:** Zegao Wei, Chao Yan, Lixian Cui, Xudong Zhao, Liang Liu

**Affiliations:** 1Department of Psychosomatic Medicine, Shanghai East Hospital, School of Medicine, Tongji University, Shanghai 200092, China; 1911689@tongji.edu.cn; 2School of Psychology and Cognitive Science, East China Normal University, Shanghai 200062, China; cyan@psy.ecnu.edu.cn; 3Division of Arts and Sciences, Center for Global Health Equity, NYU Shanghai, Shanghai 200124, China; lc145@nyu.edu; 4Clinical Research Center for Mental Disorders, Shanghai Pudong New Area Mental Health Center, School of Medicine, Tongji University, Shanghai 200092, China; zhaoxd@tongji.edu.cn

**Keywords:** adolescent depression, parental conflict discussion, physiological and facial expression response, biomarkers

## Abstract

Adolescents are a vulnerable population with a high prevalence of depression, yet there is a scarcity of biological markers for diagnosing depression specifically in this age group. In this case–control study, we examined physiological responses and facial expressions in adolescents with depression compared to healthy controls during parental conflict to identify potential biomarkers for adolescent depression. We recruited 33 families with adolescents diagnosed with depression and 25 families with healthy adolescents, matched for gender, age, and education. Baseline physiological measures, including electrocardiography (ECG), electrodermal activity (EDA), and respiration (RESP). During a 30 min parental conflict discussion, recorded on video, we analyzed the adolescents’ responses. The major depressive disorder (MDD) group displayed higher baseline heart rate (HR) and lower respiratory sinus arrhythmia (RSA). During the conflict discussion, they showed increased HR and shorter tonic periods of EDA compared to the healthy group. Facial expressions of both groups included neutral, sad, angry, and surprised. The MDD group exhibited fewer happy expressions. Receiver operating characteristic (ROC) curve analysis indicated that HR, interbeat interval (IBI), average NN interval (AVNN), number of NN50 intervals (NN50), and percentage of NN50 intervals (pNN50) had diagnostic potential for adolescent depression, with an area under the curve (AUC) greater than 0.7. Our findings suggest that adolescents with depression experience heightened sympathetic activation (higher HR) and weakened parasympathetic activity (lower RSA and HRV). These biomarkers hold promise for diagnosing adolescent depression.

## 1. Introduction

Adolescent depression is mainly characterized by clinical manifestations such as depressed mood, lack of pleasure, and physical fatigue [1]. Studies have shown that the prevalence of mood disorders increases significantly during adolescence and that the prevalence of depression is almost twice as high in girls as in boys [2,3]. Adolescents suffering from depression may be suicidal, have difficulty in emotional processing and interpersonal relationship, and refuse to go to school, among many other problems. Affective disorders during adolescence have a clear tendency to relapse and tend to be chronic, with residual symptoms such as irritability, depressed mood, and somatic discomfort. For adolescents with depression, their social functioning is more difficult to return to pre-morbid levels, even leading to the propagation of such difficulties across the lifespan [4].

As one of the most crucial emotional support systems for adolescents, the family often plays a pivotal role in adolescents’ physical and mental health. Studies have shown that parental cognitive styles, parental depression levels, parental marital conflict, and family members’ attachment styles all have an impact on the development of adolescent depression [5]. Previous studies have shown that adolescents who witness parental conflicts are prone to internalized anxiety and depression, particularly for girls, or externalized behavioral problems, particularly among boys [6]. Adolescents’ cognitive assessment of parental conflict including three main components: perception of threat, self-blame, and coping efficacy, also play roles in adolescent health. Perception of threat is the degree to which children subjectively perceive parental conflict as a threat to themselves, self-blame is the degree to which children blame themselves for the cause of parental conflict, and coping efficacy is the child’s perceived ability to cope and help parents mitigate conflict and regulate their own emotions [7]. Through cognitive assessment, adolescent children form perceptions of parental conflict, which in turn impact their emotional security and ultimately contribute to the emergence of internalizing symptoms and emotional difficulties [8]. Based on cognitive situational theory and emotional security theory, research has shown that adolescents’ negative emotions, such as fear, vigilance, and sadness, may arise in parental conflict situations [7,9,10]. In addition, the pathogenesis of depressive disorders in adolescents related to the family environment has been studied extensively by scholars, and the previously selected indicators of psychophysiological responses of children during parental conflict include free fatty acids and cortisol in blood, heart rate, blood pressure, respiration, skin conductance, and skin temperature. But most relevant studies would use heart rate, skin conductance, and skin temperature as they are more easily detectable indicators [11] and are considered to be valid indicators for early assessment and screening of the risk of mood disorders.

However, previous studies have not clarified the physiological mechanisms by which the emotional disturbances of adolescents in parental conflict situations are getting under the skin and triggering the associated emotional responses. Because of recent advancements in scientific measurement techniques, researchers are able to measure facial expressions and physiological responses more accurately. Recent studies of sympathetic and parasympathetic nervous system functioning have been focusing on physiological indicators such as electromyography, electroencephalography, electrocardiography, pulse oximetry, blood pressure measurements, respiratory sensors, body temperature measurements, and electrical skin responses [12]. By collecting data from ECG, we can obtain the index of Heart rate variability (HRV) and respiratory sinus arrhythmia (RSA). HRV is the change in the time interval between consecutive heartbeats [13]. Usually, during inspiration, the heartbeat interval becomes shorter and the heart rate increases; during expiration, the heartbeat interval becomes longer and the heart rate slows down, a normal phenomenon called RSA [14]. In recent years, RSA has been widely used as a measure of physiological regulation in electrophysiological studies of mood disorders as a good indicator of parasympathetic nervous system function [15]. Cui, L. et al., 2015 found that RSA, obtained by capturing ECG in adolescents in conflict situations, was a good indicator of parasympathetic nervous system activation [16]. Several previous studies have suggested that RSA is a reliable marker of the degree of activation of the autonomic nervous system and processes related to emotion regulation [17,18,19,20].

In addition to micro-level indicators such as physiological reactions, what we can more easily observe in the experimental situation are the changes in facial expressions. In reality, adolescents need to gather the courage to face the tension and fear of witnessing parental conflict in their home, and cannot escape experiencing these intense and complex feelings. The maturity of facial expression recognition technology makes it possible to capture the emotional state of a child’s facial expressions in situations of parental conflict. Facial expression recognition (FER) is used to detect facial expressions through specific features, such as lips and eyebrow shapes of human faces, and extract them through image processing technologies, such as grayscale, threshold, and edge detection, and capture different movements or positions of facial muscles to judge the current emotional state of the subject [21,22]. Previous studies of facial expression activity in depressed patients by neuromyography have shown that depressed patients have reduced facial muscle activity compared to normal individuals, suggesting that neutral and sad facial expressions occur more frequently in individuals with depressive symptoms [23]. In addition, it has been suggested that when the severity of depressive symptoms is higher, participants have reduced smile responses [24,25,26], tighter corners of the mouth [24], more frowning [26], shorter transient intervals [25] and more facial expressions related to contempt [27]. This makes it possible to explore the facial expression characteristics exhibited by adolescents with depression when they are faced with parental conflict situations.

Therefore, this study aimed to examine the physiological and psychological responses of adolescents in both healthy and depressed groups during instances of parental conflict, with the objective of identifying potential biological markers that could aid in diagnosing depression in adolescents. Physiological indicators, such as heart rate variability (HRV), electrodermal activity (EDA), and respiratory sinus arrhythmia (RSA), were measured, while the emotional state of facial expressions in adolescents who witnessed parental conflict was also recorded.

## 2. Materials and Methods

### 2.1. Participants

This study was conducted with the approval of the Ethics Committees of Tongji University School of Medicine and East China Normal University. Data were collected in the Psychological and Cognitive Laboratory of East China Normal University. All participated parents and adolescents provided full and comprehensive informed consents and assents prior to the start of data collection.

The MDD group (N = 33; 75.76% female; average age = 15.09 years) was recruited mainly from outpatients at the Department of Clinical Psychology of Shanghai East Hospital and Shanghai Pudong Mental Health Center from 2019 to 2021. Inclusion criteria for the case group were as follows: (1) adolescents aged 13-24 years, Han Chinese, and right-handed; (2) first-episode depression diagnosed by two psychiatrists with intermediate title or higher in accordance with the DSM-5 diagnosis of major depressive disorder; (3) exclusion of organic brain disease and other mental disorder diagnoses and comorbidities; and (4) voluntary participation of the adolescents and support and cooperation of both parents.

The HC group (N = 25; 48% female; average age = 15.16 years) was recruited from school public lectures and community networks. Inclusion criteria for the healthy group were as follows: (1) adolescents aged 13–24 years, Han Chinese, and right-handed; (2) the result of Mini International Neuropsychiatric Interview (MINI) was negative, administered face-to-face by uniformly trained researchers; (3) both 17-item Hamilton Rating Scale for Depression (HAMD-17) and Hamilton Anxiety Rating Scale (HAMA) scores were less than 7; (4) exclusion of other somatic and organic brain disorders; and (5) voluntary participation of the adolescents and support and cooperation of both parents.

There were no significant differences between the two groups in terms of gender, age, education level, family economic status, height, and weight (Table 1). The healthy control group was enrolled following the principle of attempting to match the MDD group in terms of gender, age, and education level. Moreover, according to research in adolescent topics and neuroscience, some scholars argue that young adults have not yet completed their physiological and social identity transitions [28], and the prefrontal cortex of the brain is still developing [29]. To better reflect the actual psychological development and brain maturation of adolescents, the age range of adolescence could be extended to 24 years [28,29]. In our study, we considered the cultural context in China, where individuals under the age of 24 often still rely on their parents for emotional and material support and are influenced by the quality of their parents’ relationship. Therefore, we defined the adolescent age range in our study as 13–24 years and included these young adults as participants.

Additionally, we focused on recruiting participants who belonged to the Han Chinese and were right-handed. This choice was made due to China’s multiethnic culture, with the Han population representing the majority. By excluding participants from other ethnic backgrounds, we aimed to maintain a consistent cultural context among the included families, reducing potential confounding factors related to culture. All the adolescent participants in our study were right-handed, which allowed for better matching of physiological and organic factors between the two groups, and independent of disease diagnosis. The prevalence of right-handed individuals in the population made recruitment and sampling more convenient, while also increasing the sample’s homogeneity within the depressed and healthy group.

### 2.2. Face-to-Face Assessment and Questionnaire Evaluation

Upon arriving at the assessment laboratory, both groups of adolescents underwent a face-to-face interview in a separate room, which lasted approximately 20 min. The assessment was conducted by two uniformly trained research assistants who evaluated the adolescents’ depression and anxiety levels using the Hamilton Depression Scale-17 items and the Hamilton Anxiety Scale. Furthermore, the healthy control group underwent a face-to-face MINI assessment to eliminate the possibility of other psychiatric disorders. The inter-rater correlation coefficients for HAMD, HAMA, and MINI were all greater than 0.8. Self-administered demographic questionnaires were used to collect sociodemographic information, including gender, age, years of education, height, weight, and family economic status. To assess the adolescents’ mental health, the Patient Health Questionnaire-9 (PHQ-9) and Generalized Anxiety Disorder-7 (GAD-7) were used for self-assessment of depression and anxiety, respectively. Family functioning was evaluated using the Family Assessment Device (FAD) [30] and the Self-Rating Scale of Systemic Family Dynamics (SSFD) [31]. The Self-Esteem Scale (SES) was used to assess self-esteem levels, and the Connor-Davidson Resilience scale (CD-RISC) was used to evaluate the resilience of these adolescents. Furthermore, the Childhood Trauma Questionnaire (CTQ) was used to assess childhood trauma, and the Adolescent Self-Rating Life Events Check List (ASLEC) was used to evaluate adolescent life events.

These scales are commonly used in clinical assessments for Chinese adolescents, with good reliability and validity (Due to the word limit, a detailed description of the psychometric quality of the questionnaires in the Chinese population can be found in the Appendix A).

### 2.3. Parental Conflict Situation Procedure

#### 2.3.1. Experimental Setup of Parental Conflict Situations

Both parents were invited to discuss 3 conflict topics that they had not resolved in their daily lives in a study room for 30 min. The specific conflict topics were discussed by the couple in a separate, quiet study room for 15 min prior to the formal discussion and were kept confidential from the adolescents until the formal discussion. The parents were seated in two single chairs facing the same direction, with their respective bodies turned slightly to the side at a distance of about 45° to ensure that they were in an appropriate communication position and could clearly see each other’s facial expressions [32]. At this time, the adolescent subjects sat in the middle of the two parents directly across from each other at a distance of 1.8 m to observe the parents’ discussion. The adolescents did not make any interruptions or interfere with the discussion [33]. The researcher left the observation room after verbal instruction [11]. The entire discussion process was audio-video recorded.

#### 2.3.2. Acquisition of Physiological Data at the Baseline and during the Conflict

The researchers used BIOPAC MP160 to collect EDA, ECG, and RESP data. Prior to the conflict discussion, adolescents sat in a separate, quiet study room with moderate light and a room temperature of 24–28 °C while physiological data were simultaneously collected. The 10 min data served as a baseline reference for assessing changes in these indicators during the 30 min parental discussion later, when physiological data were continuously collected from the adolescents as they watched the parental discussion [16]. This enabled the researchers to obtain the changes in the adolescents’ autonomic responses and patterns as the parental discussion progressed.

#### 2.3.3. Facial Expressions Capture and Detection

During the 30 min parental conflict situation, the facial expressions of the adolescents were recorded using the Observer XT and Media Recorder. The laboratory camera used was a dual-camera system from AXIS (P5515-E PTZ), with one camera capturing the overall interaction of the family, and the other camera specifically capturing the frontal facial expressions of the adolescents, to obtain 30 min of facial expression emotional responses from the adolescents.

### 2.4. Data Processing and Statistical Analysis

The data collected and preprocessed in this study were analyzed using HRV Analysis 3.1, EDA Analysis 3.0, and SPSS 20.0. Mindware HRV Analysis, and EDA Analysis were used to process the electro-physiological data, including heart rate, respiration sinus arrhythmia, inter-beat interval, and respiration rate. These physiological indices were recorded in real-time and were analyzed using SPSS 20.0 for subsequent data analysis. Facial expression analysis was performed using the Face Reader 7.0 software from Noldus. This software uses 500 key muscle points to create a facial model and analyze facial expressions. Six basic facial expressions, including happiness, sadness, anger, surprise, fear, disgust, and contempt, can be analyzed by using this software. The software performs facial recognition through three steps, including facial detection, facial modeling, and facial expression classification, to obtain preliminary data on facial expressions [34,35].

Subsequent data analysis was conducted using SPSS 20.0. The chi-square test or Fisher’s exact probability test was used to compare categorical variables. Normally distributed continuous variables were described using mean ± standard deviation, and the two independent sample *t*-test was used for group comparisons. Pearson correlation analysis was used for normally distributed continuous variables. All statistical analyses were two-sided, and a *p*-value less than 0.05 was considered statistically significant. To explore potential diagnostic biomarkers for adolescent depression, this study utilized the ROC curve (receiver operating characteristic curve), a commonly employed tool in medical diagnosis. The AUC (area under the curve) was used to evaluate the diagnostic efficacy of the biomarkers. An AUC value greater than 0.7 indicates acceptable diagnostic significance. The ROC curves were generated using SPSS 20.0.

## 3. Results

### 3.1. Risk and Protective Factors Associated with Depression in Two Groups of Adolescents

There were no significant differences in the basic sociodemographic characteristics between the two groups of participants (Table 1). However, significant differences were observed between the depressed and healthy groups in terms of psychological state, family functioning, and life events, as indicated by the results of the questionnaires (Table 2). Specifically, the MDD group had lower scores on measures of family system logic, suggesting that they may have more polarized thinking and less ability to accommodate differences compared to healthy families. In addition, they had lower scores on measures of family functioning, including widely used family functioning scales and a domestically developed SSFD scale, indicating that their families may have persistently poor functioning. Regarding psychological resilience, the MDD group had significantly lower scores than healthy adolescents in terms of resilience, especially in terms of toughness, ability to tolerate negative changes, and acceptance of positive changes. Furthermore, the MDD group were more likely to have experienced childhood trauma such as emotional abuse, physical abuse, and emotional neglect, and were more likely to experience stress from life events and interpersonal relationships during adolescence.

### 3.2. Comparison of the Characteristics of Physiological and Facial Expressions between MDD and HC

The differences in electrophysiological responses between the two groups of adolescents were as follows (Table 3). In a laboratory, the MDD group had higher HR, lower RSA, and lower HRV (IBI, SDNN, AVNN, RMSSD, NN50, and pNN50) levels in the baseline. This suggests that even in a calm state, the MDD group have higher sympathetic nervous system activation (higher HR) and lower parasympathetic nervous system activity (lower RSA and lower HRV), potentially indicating autonomic dysfunction. In the conflict situation, patients group also had higher HR and lower HRV, with a shorter tonic period in skin conductance activity.

In the context of witnessing parental conflict, adolescents in the depression group displayed fewer happy emotions and more neutral, sad, and contemptuous emotions, while healthy adolescents showed more happy emotions (Table 4), possibly due to the use of humor by parents in their communication. It should be noted that some subjects adopted avoidance behaviors when witnessing parental conflict, such as wearing hats or leaving their hair down to cover their faces, constantly bowing their heads, or closing their eyes, resulting in the failure to capture their facial expressions by the software. Therefore, we only included the subjects whose facial expressions were successfully captured by the software for analysis.

The results indicate that SDNN, AVNN, RMSSD, NN50, pNN50, and RSA at baseline are strongly positively correlated with each other (Table 5) and also show a strong positive correlation with IBI. This means that higher levels of HRV and RSA can both reflect better regulation of the autonomic nervous system (ANS), and there may be a possibility of mutual regulation between the two.

### 3.3. Exploration of Possible Biomarkers of Depression in Adolescents Using ROC Curves

Receiver operating characteristic (ROC) curve analysis was employed to evaluate the diagnostic value of the biomarkers for adolescent depression. Figure 1 and Figure 2 illustrate the ROC curves of the physiological indices in the sample, aiming to differentiate between the MDD group and the healthy group.

During the baseline period, the area under the curve (AUC) for the indices of B_Mean Heart Rate and B_Respiration Rate were 0.726 and 0.554, respectively (Figure 1a). The AUC values for B_RSA, B_Mean IBI, B_SDNN, B_AVNN, B_RMSSD, B_NN50, and B_pNN50 were 0.661, 0.727, 0.719, 0.727, 0.696, 0.717, and 0.718, respectively (Figure 1b). In both the baseline and conflict periods, the AUC values for the skin conductance indices B_Tonic SCL, B_Mean SC, C_Tonic SCL, C_Mean SC (Figure 1c), B_Tonic Period, and C_Tonic Period (Figure 1d) were 0.541, 0.538, 0.527, 0.53, 0.562, and 0.668, respectively.

During the conflict period, the area under the curve (AUC) values for the indices of C_Respiration Rate, C_RSA, C_Mean IBI, C_SDNN, C_AVNN, C_RMSSD, C_NN50, and C_pNN50 were 0.513, 0.673, 0.742, 0.686, 0.742, 0.675, 0.7, and 0.709, respectively (Figure 2a,b). Compared to the physiological indices during the baseline period, the arousal level indices induced by the conflict situation, including Arousal_RSA, Arousal_Resp, Arousal_AVNN, Arousal_TonicSCL, Arousal_MeanSC, and Arousal_TP, had AUC values of 0.59, 0.545, 0.507, 0.537, 0.541, and 0.516, respectively (Figure 2c). Additionally, the AUC values for Arousal_HR, Arousal_IBI, Arousal_SDNN, Arous-al_RMSSD, Arousal_NN50, and Arousal_pNN50 were 0.549, 0.507, 0.602, 0.61, 0.581, and 0.594, respectively (Figure 2d).

It can be observed that the mean HR in the baseline period has some representativeness, with an area under the curve of 0.726. The AUC values of HRV (IBI, AVNN, NN50, and pNN50) biomarkers were all greater than 0.7 for both baseline and conflict periods, indicating that these biomarkers may have potential diagnostic significance for adolescent depression.

## 4. Discussion

This is a study that reflects real-life family interactions. Continuous physiological recordings and facial expression capture provided conditions for more sensitive detection of the physiological and emotional impacts of parental conflict on their child. Additionally, also enabled the possibility of discovering physiological differences between depressed and healthy adolescents.

The main results of this study are as follows: (1) Self-esteem, psychological resilience, good family functioning, and more stable physiological states (higher RSA and higher HRV) were protective factors for adolescent depression, while childhood trauma and negative life events were risk factors for the depressed group. (2) Even in the baseline calm state, the MDD group manifested higher HR, lower RSA, and lower HRV. During parental conflict situations, their HRV showed similar patterns to baseline, and their skin conductance level decreased during the tonic period, indicating lower emotional stability than the healthy group. (3) The AUC values of HRV (IBI, AVNN, NN50, and pNN50) biomarkers were all greater than 0.7 for both baseline and conflict periods, indicating that these biomarkers may have potential diagnostic significance for adolescent depression. (4) As for the facial expressions in parental conflict situations, both groups of adolescents exhibited negative emotional experiences, primarily neutral, sad, surprised, and angry, when witnessing parental conflict, with the depressed group exhibiting a lower frequency of experiencing happy emotions than the healthy group.

Depression as a complex psychological disorder has been extensively researched in terms of its risk and protective factors. The findings of this study are consistent with previous research in the field. Specifically, the vulnerability model, which posits that low self-esteem is a risk factor for depression [36], is supported by the results of this study. Additionally, Palosaari’s 1995 study suggested that while low self-esteem and an adverse family environment during adolescence are risk factors for depression, intimate relationships during youth can still serve as a protective factor against depression, regardless of gender [37]. Other related research also highlights the importance of psychological resilience as a protective factor against adolescent depression and the role of self-compassion skills in helping adolescents cope with challenging life circumstances [38]. Family cohesion and psychological resilience have also been found to reduce the occurrence of suicidal ideation in adolescents [39]. As such, psychological therapy for adolescent depression can focus on improving self-esteem, enhancing psychological resilience, and strengthening family functioning to reinforce protective factors and reduce risk factors. It is important to note that negative life events, such as childhood trauma and adverse adolescent life events, have been shown to be strong risk factors for depression [40,41,42]. Therefore, parents and families have a crucial role in providing necessary protection and a safe environment for children and adolescents, in order to minimize the occurrence of such events to the greatest extent possible.

The emotional reactions of children and adolescents during parental conflicts have been extensively researched by scholars [43,44,45,46,47]. However, the real-time capture of facial expressions in young people is a novel and innovative approach that has only recently become possible through technological advancements. Notably, the present study faced some challenges in collecting facial expression data due to potential confounding factors. For instance, some adolescents may have deliberately obscured the upper part of their faces using hats or hair to avoid facing their parents during conflicts. As a result, genuine facial expressions may have been missed, and the system could have recorded them as “unknown.” Additionally, ethical considerations compelled the researchers to avoid imposing strict requirements for the participants to face the camera, allowing them some degree of freedom to capture more natural and genuine reactions during conflicts with their parents. The findings of this study indicate that both the depressed and healthy groups exhibited negative emotions, such as sadness, disgust, fear, and contempt during conflicts with their parents, while the healthy group exhibited more happy emotions. The difference in emotional response between the two groups was statistically significant. This could be attributed to parents in the healthy group’s use of humor and interesting expressions during conflict interactions, as opposed to parents in the depression group, who may have exhibited caution and contributed to a relatively suppressed and dull family atmosphere during discussions. However, given that adolescents in the depression group already exhibit reduced levels of positive emotions, it is possible that this could be a clinical feature of the disorder itself. In future research, it may be prudent to incorporate video analysis of parental conflict interaction styles with real-time correlation analysis between negative or positive facial expressions in children and adolescents to achieve more comprehensive and informative results.

However, the main focus of this study is to compare the differences in electrophysiological responses between adolescents with depression and healthy controls. The results indicate that adolescents with depression show dysregulation in the autonomic nervous system, which differs significantly from healthy adolescents. A case–control study published by Koschke et al. in 2009 [48], which assessed heart rate variability, QT variability, and baroreflex sensitivity in 75 patients with major depressive disorder (MDD), also suggested that MDD can cause dysregulation in the autonomic nervous system, particularly with increased sympathetic activity, and that this trend may be exacerbated to some extent by treatment with serotonin norepinephrine reuptake inhibitors (SNRIs) and selective serotonin reuptake inhibitors (SSRIs). Similar findings [49] have been observed in studies using rodents as research subjects, with dynamic electrocardiogram results in mice indicating increased HR and decreased HRV during depressive-like states. These findings suggest that depressive-like behavior in both mice and humans is often accompanied by changes in autonomic nervous system function. Furthermore, more studies have shown that the effects of depression on autonomic nervous system function may have far-reaching consequences on the occurrence of cardiovascular diseases, such as coronary heart disease and hypertension, in patients in the future [50]. Changes in autonomic nervous system function are considered to be one of the pathophysiological mechanisms linking psychological factors, such as depression and anxiety, to cardiovascular diseases [51]. A study by Sgoifo et al. in 2015 also suggested that decreased HRV may manifest as emotional dysregulation, decreased psychological flexibility, and impaired social engagement and that this indicator may serve as a specific measure for evaluating and preventing the development of psychiatric disorders [52], indicating that exploring physiological indicators may help identify potential biomarkers with diagnostic significance for adolescent depression [53]. The findings from these studies are consistent with the results of our study, and we also hope that further exploration of specific physiological indicators may lead to the discovery of new potential therapeutic targets for the diagnosis and treatment of adolescent depression.

Depression is a prevalent cause of suicide in adolescents, with a higher incidence during this developmental stage. However, due to their unique psychological state, adolescents may display emotions of “anger” and “hostility” during diagnosis. Furthermore, the current diagnostic system for depression in psychiatry relies mainly on subjective information collected through interviews and questionnaires, which may lead to misdiagnosis and underdiagnoses. This highlights the need for objective biological markers in the diagnostic process. Research on biomarkers for depression has focused on various systems, such as inflammation, growth factors, neurotransmitters, endocrine and metabolic factors, neuroimaging markers, and genetics [54]. However, practical biomarkers for clinical application have yet to be developed, and research on exploring the biological markers of depression through electrophysiology is still relatively limited. Thus, this study primarily explores potential diagnostic biomarkers for adolescent depression at the electrophysiological level. The results indicate that HR, IBI, AVNN, NN50, and pNN50 could serve as auxiliary diagnostic biomarkers for depression, as they show good classification performance in both baseline and conflict periods. However, more research is necessary to determine whether these indices can indicate the severity of adolescent depression or the occurrence, development, and changes in the disease. Unfortunately, anxiety symptoms often coexist with depression in adolescents, which may affect the physiological changes observed. Future studies could optimize the purity of the samples and collect longitudinal electrophysiological data to clarify the representativeness and specificity of these biomarkers in the physiological characteristics of adolescent depression. This will provide more effective support and evidence for future clinical applications. Moreover, to gain further insights into the relationship between a child’s electrophysiological activity and parental conflict behavior, new data analysis methods such as time series analysis should be employed. This may help to identify specific changes in adolescent electrophysiological activity during parental conflict situations and their relationship with the interaction behavior of parents.

## 5. Limitations

This study has several limitations. Firstly, it is a small-sample study. Adolescent depression within families represents a relatively rare sample, and the involvement of family members and the simulation of real-life contexts in this study posed challenges during the recruitment phase. The limited sample size restricts more in-depth research exploration and analysis. Hence, we should be circumspect when extrapolating the results to larger populations.

Secondly, this study attempted to simulate real-life situations within a laboratory setting. However, despite providing informed consent and prior notification, the audio-video recordings in the laboratory environment may still have interfered with the natural dynamics of parental conflict discussions. This interference may have also affected adolescents’ natural display of genuine emotions and facial expressions. Thirdly, as mentioned in the discussion section, some adolescents exhibited avoidance behaviors during parental conflicts, such as covering their faces with hats or hair, which limited the collection of facial expression data. Consequently, certain facial expressions were unable to be captured, modeled, or classified.

However, our study represents an initial exploration and attempt to simulate parental conflict situations in real-life contexts and capture children’s physiological responses and facial expressions. We hope that in the future, with the rapid development of artificial intelligence and wearable devices, this research can extend beyond the laboratory and be applied in a broader range of real-life and clinical applications.

## 6. Conclusions

Witnessing parental conflicts can bring negative emotional experiences to adolescents. The MDD group is more likely to show autonomic dysfunction, manifested by stronger sympathetic excitation (higher HR) and weaker parasympathetic activation (lower RSA and lower HRV). The AUC (area under the curve) values of mean HR and HRV (IBI, AVNN, NN50, and pNN50) biomarkers were all greater than 0.7 for both baseline and conflict periods, indicating that these biomarkers may have potential diagnostic significance for adolescent depression.

## Figures and Tables

**Figure 1 children-10-01195-f001:**
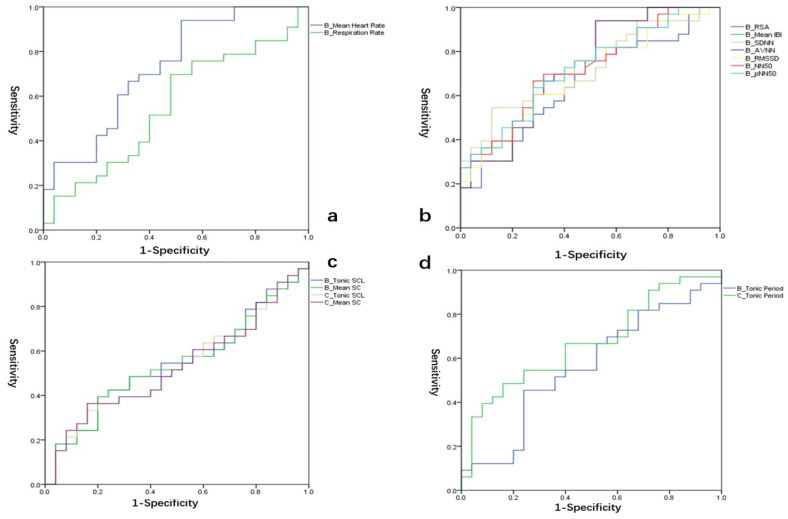
ROC curves of heart rate variability(**a**,**b**) and skin electrical activity(**c**,**d**) index at baseline.

**Figure 2 children-10-01195-f002:**
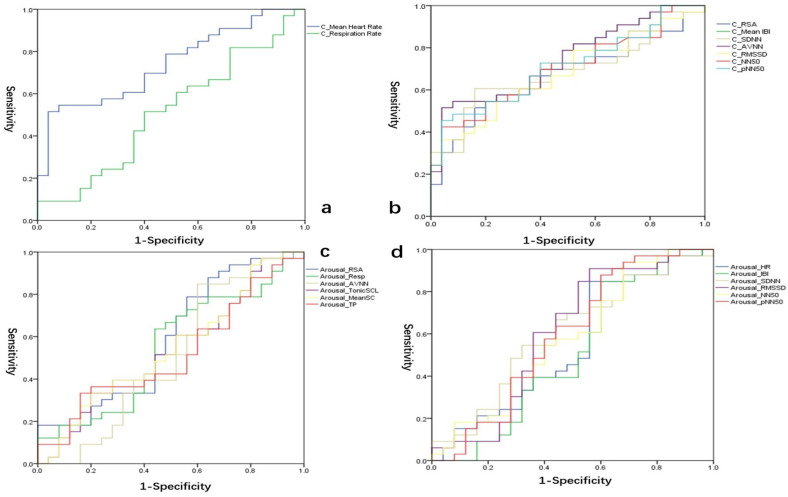
ROC curves of heart rate variability (**a**,**b**,**d**) and skin electrical activity (**c**) during conflict situations.

**Table 1 children-10-01195-t001:** Comparison of sociodemographic characteristics between adolescents with major depressive disorder (MDD) and healthy control (HC) adolescents.

Variable	MDD (n = 33)	HC (n = 25)	F/χ2	*p*
Gender				0.052
Female	25	12		
Male	8	13		
Age	15.09 ± 1.74	15.16 ± 2.82	7.09	0.91
Education Duration	9.11 ± 1.78	9.24 ± 2.69	6.32	0.83
Height/cm	165.48 ± 6.87	166.84 ± 9.21	2.59	0.52
Weight/kg	59.06 ± 12.65	60.89 ± 17.98	1.30	0.65
HAMD ^1^	21.14 ± 9.33	2.24 ± 3.06	16.91	<0.001
HAMA ^2^	21.53 ± 9.44	2.6 ± 3.39	10.85	<0.001
SES in China ^3^			7.68	0.26
2	1	0		
3	5	0		
4	5	4		
5	10	7		
6	8	8		
7	3	6		
8	1	0		

^1^ HAMD = Hamilton Rating Scale for Depression, ^2^ HAMA = Hamilton Rating Scale for Anxiety, ^3^ SES in China = Socioeconomic status in China.

**Table 2 children-10-01195-t002:** Mental status, family functioning, and life events of adolescents in the depressed and healthy groups.

Variable	MDD (n = 33)	HC (n = 25)	F/χ2	*p*
PHQ9	17.15 ± 5.93	3.32 ± 3.50	6.97	<0.001
GAD7	11.64 ± 5.81	2.80 ± 3.11	19.01	<0.001
SES	20.91 ± 4.99	31.32 ± 6.01	0.76	<0.001
ASLEC	28.64 ± 7.67	22.96 ± 6.54	1.28	<0.01
Interpersonal Stress	13.58 ± 0.85	10.08 ± 0.67	7.05	<0.01
Learning Stress	15.06 ± 0.76	12.88 ± 0.94	0.02	0.078
CD_RISC	37.09 ± 12.78	61.80 ± 16.14	0.27	<0.001
Resilience	11.18 ± 0.91	19.32 ± 1.20	0.43	<0.001
Tolerating Negative Emotions	10.00 ± 0.61	16.92 ± 0.92	0.15	<0.001
Accepting Positive Change	7.52 ± 0.53	14.24 ± 0.71	0.48	<0.001
Psychological Impact	4.06 ± 0.30	4.56 ± 0.31	0.35	0.258
CTQ	47.44 ± 15.02	36.38 ± 12.29	3.95	<0.01
Emotional Abuse	12.38 ± 0.83	7.48 ± 0.60	11.14	<0.001
Physical Abuse	8.52 ± 0.73	6.28 ± 0.51	6.39	<0.05
Sexual Abuse	6.42 ± 0.33	5.88 ± 0.22	6.47	0.206
Emotional Neglect	12.55 ± 0.76	9.88 ± 1.06	0.05	<0.05
Physical Neglect	7.61 ± 0.52	6.56 ± 0.54	0.61	0.175
FAD	32.45 ± 6.34	37.00 ± 5.25	1.73	<0.01
SSFD	75.13 ± 14.32	83.08 ± 13.31	0.90	<0.05
FA_RS	24.91 ± 1.18	27.68 ± 1.38	0.12	0.130
IN_RS	20.7 ± 0.76	21.96 ± 0.97	0.13	0.305
SL_RS	16.66 ± 0.73	19.84 ± 0.53	4.02	<0.01
DO_RS	13.39 ± 0.31	13.60 ± 0.61	5.06	0.749

Note: PHQ-9 = Patient Health Questionnaire-9; GAD-7 = Generalized Anxiety Disorder-7; FAD = Family Assessment Device; SSFD = Self-Rating Scale of Systemic Family Dynamics; SES = Self-Esteem Scale; CD-RISC = Connor-Davidson Resilience Scale; CTQ = Childhood Trauma Questionnaire; ASLEC = Adolescent Self-Rating Life Events Check List.

**Table 3 children-10-01195-t003:** Physiological characteristics between MDD adolescents and HC adolescents.

Variable	MDD (n = 33)	HC (n = 25)	F/χ2	*p*
B_Mean Heart Rate	81.93 ± 1.90	73.08 ± 1.85	0.70	<0.001
B_RSA	5.89 ± 0.22	6.62 ± 0.19	0.98	<0.05
B_Mean IBI	745.10 ± 16.06	835.89 ± 21.86	0.45	<0.001
B_Respiration Rate	16.81 ± 0.56	16.27 ± 0.6	0.00	0.512
B_SDNN	52.03 ± 4.11	72.95 ± 5.62	0.59	<0.001
B_AVNN	745.10 ± 16.06	835.89 ± 21.86	0.45	<0.001
B_RMSSD	42.04 ± 4.70	61.21 ± 6.43	1.13	<0.05
B_NN50	11.89 ± 1.98	21.09 ± 2.46	0.05	<0.01
B_pNN50	16.69 ± 2.87	31.52 ± 3.99	0.71	<0.01
B_Tonic SCL	84.12 ± 11.25	80.08 ± 17.59	0.00	0.847
B_Mean SC	85.50 ± 11.50	80.77 ± 17.75	0.01	0.824
B_Tonic Period	41.77 ± 1.57	43.84 ± 1.62	0.20	0.371
C_Mean Heart Rate	85.49 ± 1.91	75.95 ± 1.69	2.69	<0.001
C_RSA	5.81 ± 0.21	6.33 ± 0.17	1.95	0.068
C_Mean IBI	714.04 ± 15.67	802.85 ± 19.46	0.05	<0.001
C_Respiration Rate	17.71 ± 0.48	17.53 ± 0.54	0.00	0.812
C_SDNN	54.47 ± 4.21	69.24 ± 4.75	0.01	<0.05
C_AVNN	714.04 ± 15.67	802.85 ± 19.46	0.05	<0.001
C_RMSSD	42.81 ± 4.83	54.06 ± 5.12	0.02	0.116
C_NN50	11.04 ± 1.84	17.80 ± 2.20	0.11	<0.05
C_pNN50	14.66 ± 2.61	26.03 ± 3.71	1.21	<0.05
C_Tonic SCL	137.75 ± 14.12	145.61 ± 27.40	0.08	0.786
C_Mean SC	139.88 ± 14.36	147.77 ± 27.68	0.07	0.788
C_Tonic Period	32.82 ± 1.08	36.56 ± 1.16	0.27	<0.05

Note: B means baseline; C means conflict; RSA = respiratory sinus arrhythmia; IBI = interbeat interval; AVNN = average NN interval; NN50 = number of NN50 intervals; pNN50 = percentage of NN50 intervals; SCL = skin conductance Level; SC = skin conductance.

**Table 4 children-10-01195-t004:** Comparative analysis of facial expression states in two groups of subjects during parental conflict.

**Variable**	**MDD (n = 18)**	**HC (n = 19)**	**F/χ2**	** *p* **
Gender				0.313
Female	13	10		
Male	5	9		
SES in China			7.88	0.25
2	1	0		
3	4	0		
4	1	2		
5	4	7		
6	5	6		
7	2	4		
8	1	0		
AGE	14.72 ± 0.30	15.32 ± 0.66	7.38	0.426
Education Duration	8.61 ± 0.33	9.42 ± 0.63	6.22	0.268
Height/cm	164.39 ± 1.67	166.74 ± 2.24	2.33	0.410
Weight/kg	62.44 ± 3.47	55.97 ± 2.95	0.30	0.163
HAMA	22.11 ± 2.54	2.21 ± 0.79	7.18	*p* < 0.001
HAMD	23.06 ± 2.33	1.68 ± 0.61	15.15	*p* < 0.001
PHQ-9	17.28 ± 1.45	2.42 ± 0.59	13.71	*p* < 0.001
GAD-7	11.61 ± 1.46	2.68 ± 0.73	17.40	*p* < 0.001
Neutral	78.94 ± 10.03	72.21 ± 9.47	0.28	0.628
Happy	7.06 ± 2.04	28.58 ± 7.26	17.59	*p* < 0.01
Sad	23.22 ± 6.91	21.47 ± 5.03	0.64	0.838
Angry	13.61 ± 5.52	4.11 ± 1.74	14.53	0.102
Surprise	14.94 ± 5.43	8.74 ± 2.53	7.05	0.299
Fear	2.72 ± 2.04	0.32 ± 0.32	3.87	0.239
Disgust	6.33 ± 1.96	6.37 ± 3.93	0.95	0.994
Contempt	2.28 ± 1.32	0.21 ± 0.21	10.13	0.120
Unknown	45.67 ± 6.77	40.37 ± 6.63	0.00	0.580

Note: “Unknown” indicates facial modeling failure, where emotions cannot be classified.

**Table 5 children-10-01195-t005:** Correlation analysis of adolescent depression and anxiety levels with heart rate variability at baseline state.

	1	2	3	4	5	6	7	8	9	10	11	12	13
1.HAMD	1												
2.HAMA	**0.839 ****	1											
3.PHQ9	**0.816 ****	**0.835 ****	1										
4.GAD7	**0.754 ****	**0.777 ****	**0.885 ****	1									
5.B_Mean Heart Rate	0.251	**0.261 ***	**0.322 ***	0.101	1								
6.B_RSA	**−0.329 ***	−0.223	−0.250	−0.119	**−0.660 ****	1							
7.B_Mean IBI	**−0.286 ***	**−0.289 ***	**−0.340 ****	−0.136	**−0.980 ****	**0.659 ****	1						
8.B_Respiration Rate	0.154	0.093	0.058	0.004	0.164	**−0.522 ****	−0.157	1					
9.B_SDNN	**−0.358 ****	−0.241	**−0.273 ***	−0.177	**−0.567 ****	**0.848 ****	**0.606 ****	**−0.443 ****	1				
10.B_AVNN	**−0.286 ***	**−0.289 ***	**−0.340 ****	−0.136	**−0.980 ****	**0.659 ****	**1.000 ****	−0.157	**0.606 ****	1			
11.B_RMSSD	**−0.287 ***	−0.203	−0.219	−0.096	**−0.609 ****	**0.856 ****	**0.644 ****	−0.277 *	**0.879 ****	**0.644 ****	1		
12.B_NN50	**−0.360 ****	**−0.376 ****	**−0.353 ****	−0.218	**−0.601 ****	**0.799 ****	**0.608 ****	−0.160	**0.754 ****	**0.608 ****	**0.851 ****	1	
13.B_pNN50	**−0.368 ****	**−0.380 ****	**−0.369 ****	−0.221	**−0.674 ****	**0.798 ****	**0.704 ****	−0.150	**0.772 ****	**0.704 ****	**0.868 ****	**0.983 ****	1
N	58.00	58.00	58.00	58.00	58.00	58.00	58.00	58.00	58.00	58.00	58.00	58.00	58.00
M	12.99	13.37	11.19	7.83	78.12	6.21	784.23	16.58	61.05	784.23	50.30	15.86	23.08
SD	11.91	12.01	8.52	6.52	11.06	1.20	108.94	3.10	27.46	108.94	30.59	12.54	19.37
Min	0.00	0.00	0.00	0.00	54.25	2.60	566.48	9.83	18.94	566.48	10.09	0.00	0.00
Max	47.00	48.00	26.00	20.00	106.13	8.38	1113.23	26.45	152.72	1113.23	139.15	42.90	71.07

Note: The significance of bold emphases in the table is *p* < 0.05. * *p* < 0.05; ** *p* < 0.01.

## Data Availability

Data will be available from the corresponding author upon reasonable request.

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
