# Peer review of "Physiological Responses during Parental Conflicts as Potential Biomarkers for Adolescent Depression"

_children, 2023, doi:10.3390/children10071195_

Round 1

Reviewer 1 Report

The present study compares adolescents/adults (ages 13-24) with and without major depressive disorders on a range of self-report, physiological and observational tasks.  The main aim was to examine relevant biomarkers in a parental conflict situation.

The overall approach seems very worthwhile.  I found the description of the methods to lack detail and the approach to analysis quite basic and potentially problematic.

The manuscript needs substantial work and I am unclear how man of the difficulties are due to the writing and how many are fundamental problems.  The following is an overview of the main issues.

1.       There are no hypotheses or research questions.  This is not presented as a pilot study or exploratory study, so it is important to have research hypotheses or questions.

2.       The participants are identified as adolescents in the title and elsewhere, but the participants are up to age 24.  The study is therefore with adolescents and young adults.

3.       Explanations for some inclusion decisions are needed.  For example, why exclude left-handed or ambidextrous participants?

4.       Have any of the questionnaires been validated in Mandarin?  More information is needed on use of these questionnaires with Chinese participants.

5.       Some questionnaires don’t seem appropriate for the age-group selected.  For example, the CTQ asks participants to comment on traumatic events before the age of 18.  Most participants in this study are not yet 18 years of age.  The CTQ is designed to be retrospective but for these participants the situations could be happening now – or in the future.

6.       How was weight and height measured?

7.       Was there any independent measure that conflict was experienced in the Parental Conflict Situation?  It seems possible that not everyone experienced conflict given that this was not an everyday context.  Perhaps the MDD group experienced more conflict and that has influenced the overall results.

8.       There is a long series of comparisons of means with no attempt to correct for multiple comparisons.  There is a lot of potential for false positives and the analyses focus on individual variables so provide only basic information.  It would be helpful to use multivariate statistics although it may not be possible with the sample size.

Author Response

Thank you!

Reviewer 2 Report

Dear Authors,

I appreciate conducting this novel and innovative study comparing biological markers and psychological characteristics of adolescents with and without depression during parental conflict. The introduction and discussion sections were well-written, and I recommend improving the study with the minor recommendation below. 

- Please pay attention to your language about depression. "Depressed adolescents" is a label for the adolescent. Instead of "depressed adolescents," please use adolescents with depression. 

- What does "Han Chinese" mean? Please explain. 

- Why did you choose right-handed adolescents? Please explain this inclusion criteria. 

- The measurement used in the study should be explained in detail. For example, who developed, did reliability and reliability in Chinese, and how many items they have, how they score, how is the Cronbach alpha coefficient number in your study among adolescents with and without depression? Please add details. 

- What were the conflict topics? Please explain. Are they the same for all adolescents? All adolescents have conversations on the same issues with their parents because it might affect their facial expressions and biological markers. 

- How did you calculate the facial expression? You mentioned a program; is it  artificial intelligence? Please provide more information. 

- You mentioned some limitations in the discussion part. Please write your limitations separately. 

- One of the other limitations of this study is having conflict between the eyes and video recording. Please add this statement.

- What was SES, and did categories mean? Please explain. 

- References should be written according to journal rules.

I wish you success in your work. 

Author Response

Thank you!

Reviewer 3 Report

Comments

Introduction: Keeping in view, in the recent past several other studies are also been determining the physiological markers of depression, identifying the novelty of this study with reference to the cultural or social context of Chinese society needs to be demonstrated.

Method: Just like you refer to Mini International Neuropsychiatric Interview(MINI), also write in full what is “HAMD and HAMA scores”.

Please clarify, how it became possible to collect natural physiological responses keeping in view that ethically obligated to explain the rationale of the study to participants through informed consent. Moreover, if it was hidden from adolescents, how the ethical issues were addressed such as non-maleficence?

Results and Discussion: Considering that I am not an expert in psychophysiology, I hold my comments and apparently, these sections seem sound. I am just curious if the authors could add some points in terms of limitations. For instance, how these conflicts, which happened in experimental settings match with real-life scenarios. Moreover, the adverse family environment is a known risk factor for psychological distress, but how biomarkers could be useful in terms of clinical or non-clinical management of psychological responses.

Author Response

Thank you!
